# Geographic Location Affects the Bacterial Community Composition and Diversity More than Species Identity for Tropical Tree Species

**DOI:** 10.3390/plants13111565

**Published:** 2024-06-05

**Authors:** Kepeng Ji, Yaqing Wei, Guoyu Lan

**Affiliations:** 1Rubber Research Institute, Chinese Academy of Tropical Agricultural Sciences, Haikou 571101, China; hnjikepeng@163.com (K.J.); hnweiyaqing@163.com (Y.W.); 2College of Tropical Agriculture and Forestry, Hainan University, Haikou 570228, China; 3Hainan Danzhou Tropical Agro-Ecosystem National Observation and Research Station, Danzhou 571737, China

**Keywords:** *Dacrydium pectinatum*, *Vatica mangachapoi*, bacterial composition, geographic location, plant identity, environmental factors

## Abstract

Microorganisms associated with plants play a crucial role in their growth, development, and overall health. However, much remains unclear regarding the relative significance of tree species identity and spatial variation in shaping the distribution of plant bacterial communities across large tropical regions, as well as how these communities respond to environmental changes. In this study, we aimed to elucidate the characteristics of bacterial community composition in association with two rare and endangered tropical tree species, *Dacrydium pectinatum* and *Vatica mangachapoi*, across various geographical locations on Hainan Island. Our findings can be summarized as follows: (1) Significant differences existed in the bacterial composition between *D. pectinatum* and *V. mangachapoi*, as observed in the diversity of bacterial populations within the root endosphere. Plant host-related variables, such as nitrogen content, emerged as key drivers influencing leaf bacterial community compositions, underscoring the substantial impact of plant identity on bacterial composition. (2) Environmental factors associated with geographical locations, including temperature and soil pH, predominantly drove changes in both leaf and root-associated bacterial community compositions. These findings underscored the influence of geographical locations on shaping plant-associated bacterial communities. (3) Further analysis revealed that geographical locations exerted a greater influence than tree species identity on bacterial community compositions and diversity. Overall, our study underscores that environmental variables tied to geographical location primarily dictate changes in plant bacterial community composition. These insights contribute to our understanding of microbial biogeography in tropical regions and carry significant implications for the conservation of rare and endangered tropical trees.

## 1. Introduction

Microorganisms, arguably the most diverse and abundant organisms on Earth [1], inhabit both the external and internal parts of plants, collectively forming the plant microbiome, with bacteria being particularly abundant and significant [2,3]. Together, plants and microorganisms constitute a ‘plant holobiont’, engaging in continuous interaction [4,5]. The consensus is that microorganisms associated with plants profoundly influence their growth, development, and overall health [6,7]. Plant microorganisms occupy distinct microhabitats [3]. One is the phyllosphere, which mainly refers to the leaves, including epiphytic and endophytic microbial communities [8]. Additionally, root-associated compartments, including the rhizosphere, rhizoplane, and root endosphere, are crucial microenvironments [9,10]. Previous research has consistently shown that these plant microhabitats represent specific niches characterized by unique chemical and physical properties, thereby harboring diverse microbial communities [11,12,13].

The plant-associated microbiome results from intricate interactions among the plant host, microorganisms, and the environment. Numerous studies have demonstrated that the plant microbiome is influenced by factors such as plant identity, seasonal variations, geographic location, soil properties, and others [14,15,16,17]. It is evident that the plant compartment shapes plant microorganisms, a fact that cannot be disputed [18,19]. Moreover, geographic variability is widely believed to play a fundamental role in structuring microbial communities [20,21,22]. Due to significant evolutionary associations between hosts and groups of bacteria, as well as selective filtering by hosts, bacteria have host specificity [23,24]. Consequently, considerable differences in microbial community composition exist among different plant species [25,26]. Previous studies have indicated that host species identity explains a larger proportion of variation in the phyllosphere bacterial community composition of temperate tree species than geographic location does [26].

Hainan Island of China is one of the hotspots for global biodiversity research, boasting vast areas of tropical rainforest, which cover 17.3% of the island’s total area [27,28,29]. Within this rich ecosystem, *Dacrydium pectinatum* and *Vatica mangachapoi* are significant gymnosperm and angiosperm species, respectively, renowned for their rarity and endangered status. Given the importance of host–microbiome interactions for host survival, revealing the microbial community composition of endangered and rare tree species and its influencing factors may provide possibilities for utilizing microbes to enhance plant growth adaptability in the future, which is crucial for the conservation of endangered and rare tree species [30]. However, the bacterial communities associated with tropical rare and endangered *D. pectinatum* and *V. mangachapoi*, as well as how they respond to changes in environmental factors, remain unknown.

Therefore, this study quantitatively analyzed the spatial patterns and driving factors of bacterial community structures in different compartments of plants in the tropical rainforest, focusing on *D. pectinatum* and *V. mangachapoi*, using high-throughput sequencing technology. We hypothesize that host species identity explains more variations in bacterial community composition of tropical tree species compared to geographic location, as angiosperm and gymnosperm have a distant relatedness. Our objectives are threefold: (1) to identify the bacterial community present in *D. pectinatum* and *V. mangachapoi*; (2) to quantify the relative influence of bacterial community composition: host species identity and geographic location; and (3) to elucidate the environmental drivers of bacterial community composition in tropical tree species.

## 2. Materials and Methods

### 2.1. Study Site and Sampling

The sampling locations were chosen from five core distribution areas of the forests of *V. mangachapoi* and *D. pectinatum* on Hainan Island. For *V. mangachapoi*, four study sites were selected in Bawang, Diaoluo, Jianfeng, and Wanning, while for *D. pectinatum*, three study sites were chosen in Diaoluo, Jianfeng, and Wuzhi (Appendix A). Latitude, longitude, and elevation were recorded for each sample point. Mean monthly rainfall and mean monthly temperature data were obtained from the National Meteorological Information Center (http://data.cma.cn/) for further analysis on 8 September 2023.

At each sample site, three plants spaced approximately 100 m apart were chosen for analysis, resulting in three replicates. Mature and healthy leaf samples were collected from positions 2 m away from the tree trunk and 12 m above the ground in each of the four cardinal directions (i.e., north, south, east, west) and then mixed to form a single composite sample. Root samples were obtained from the same tree using the same approach (Appendix A). Additionally, a portion of the leaf and root samples was utilized for physicochemical property analysis.

To obtain epiphytic microbiota samples, microbial cells were dislodged and collected from 15–20 g of leaves and 10–15 g of roots. Leaves were submerged in a PBS solution and subjected to ultrasonication at 40 kHz for 1 min, followed by agitation on a shaker at 200 rpm for 4 min. This process was repeated three times [31,32]. Subsequently, the solution was filtered, and the obtained filter membrane was stored at −80 °C for sequencing analysis. The leaf samples, from which surface microbiota were extracted, underwent high-throughput analysis to detect endophytes. It is worth noting that for root sample extraction, rhizosphere soil was manually separated from the roots through handshaking, while the soil adhering to the roots was considered rhizoplane soil (i.e., the soil layer approximately 1 mm thick) [9,10].

### 2.2. Soil and Leaf Physical and Chemical Property Determination

Water content (WC) and organic matter (OM) were measured gravimetrically. Total nitrogen (TN) was determined using micro-Kjeldahl digestion followed by steam distillation. Total phosphorus (TP) and total potassium (TK) were assessed using NaOH digestion. Soil (or leaf) pH was measured in a leaf/water suspension (1:2.5, *w*/*w*) using a pH meter [12].

### 2.3. DNA Extraction and PCR Amplification

Microbial community genomic DNA (gDNA) was extracted from soil, root, and leaf samples using FastDNA^®^ Spin Kit for Soil (MP Biomedicals, Irvine, CA, USA) according to the manufacturer’s instructions. PCR primers 515FmodF (5′-GTGYCAGCMGCCGCGGTAA-3′) and 806RmodR (5′-GGACTACNVGGGTWTCTAAT-3′) were used to amplify the hypervariable region V4 of the bacterial 16S rRNA gene [33,34]. The PCR amplification conditions were as follows: initial denaturation at 95 °C for 3 min; followed by 35 cycles of denaturation at 95 °C for 30 s, annealing at 55 °C for 30 s, and extension at 72 °C for 45 s; with a final extension at 72 °C for 10 min. The purified amplicons were subjected to equimolar and paired-end sequencing on the Illumina MiSeq platform at Shanghai Majorbio Bio-pharm Technology Co., Ltd., Shanghai, China. The PCR mixtures contain 5 × TransStart FastPfu buffer 4 μL, 2.5 mM dNTPs 2 μL, forward primer (5 μM) 0.8 μL, reverse primer (5 μM) 0.8 μL, TransStart FastPfu DNA Polymerase 0.4 μL, template DNA 10 ng, and finally, ddH_2_O up to 20 μL. PCR reactions were performed in triplicate. The raw reads were deposited into the National Center for Biotechnology Information Sequence Read Archive database (Accession Number: PRJNA1085516).

### 2.4. Bioinformatics and Data Analysis

The raw FASTQ files were demultiplexed and quality-filtered using the Quantitative Insights Into Microbial Ecology (QIIME) microbiome ecology software (version 1.17) [35]. Non-redundant sequences (excluding singletons) were clustered into operational taxonomic units (OTUs) at a 97% similarity threshold, and chimeric sequences were identified and removed using UCHIME, resulting in effective sequences with over 97% similarity [36]. The sequences were taxonomically classified using the RDP Classifier version 2.2 [37]. Operational taxonomic units (OTUs) matching chloroplast and mitochondrial sequences were removed.

We calculated the diversity index (OTU richness, observed number of OTUs) for the grouped samples using the Vegan R (4.3.0) software package [38]. Pairwise comparisons of the results were conducted using Wilcoxon tests, with *p*-values adjusted using the false discovery rate method. Analysis of variance was performed with the diversity index as the response variable and plant compartment, species, and geographical location variation as fixed effects. The cumulative number of OTUs for five compartments was utilized to estimate γ-diversity. Principal coordinate analysis (PCoA) was employed to visualize differences in bacterial community compositions (OTU level), and group differences were assessed using permuted multivariate analysis of variance (PERMANOVA) with 999 permutations. To identify statistically different biomarkers between groups (Wilcoxon *p*-value < 0.05, logarithmic LDA score > 2), we utilized linear discriminant analysis (LDA) coupled with effect size analysis (LEfSe) [11,39].

To elucidate the relationship between the relative abundances of the dominant bacterial class and environmental factors, a linear regression model was used for evaluation. The mantel test and redundancy analysis (RDA) were used to analyze the impact of soil and leaf properties and climate factors on bacterial communities. We used the function varpart in the Vegan package to quantify the relative importance of tree species, climate factors (rainfall and temperature), soil and leaf physicochemical properties, and geographical variables [40].

## 3. Results

### 3.1. The Community Compositions

In total, paired-end sequencing resulted in 3,808,575 high-quality reads, detecting 154,445 OTUs representing the bacterial community. Both geographical location and tree species identity exerted significant influence on bacterial community compositions (Figure 1 and Appendix A). Specifically, the relative abundance of *Gammaproteobacteria* and *Actinobacteria* was notably higher in *D. pectinatum* compared to *V. mangachapoi* (Figure 1A). LDA effect size analysis (LEfSe) identified *Gammaproteobacteria* in *D. pectinatum* and *Actinobacteria* in *V. mangachapoi* as the most significant biomarkers at the taxonomic level (Appendix A). Higher relative abundances of *Gammaproteobacteria* and *Alphaproteobacteria* in *D. pectinatum* were observed across Wuzhi and JianFeng respectively, compared to other locations (Figure 1B and Appendix A). Furthermore, higher abundances of *Actinobacteria* were observed in *V. mangachapoi* in Wanning, while the relative abundances of *Gammaproteobacteria*, *Alphaproteobacteria*, and *Acidobacteriae* in *V. mangachapoi* varied significantly across different geographical locations (Figure 1C and Appendix A). Principal coordinate analysis (PCoA) and permuted multivariate analysis of variance (PERMANOVA) results indicated that plant compartment significantly influenced bacterial community composition, followed by geographical location and tree species identity (Figure 2A and Appendix A). Geographical location and tree species identity explained 4.3% and 1.4%, respectively. Further analysis revealed that geographical location had a significant impact on the five compartments than tree species identity (Figure 2B and Appendix A). Specifically, 34.3% and 47.1% of the bacterial variance in the leaf epiphytic and endophytic, respectively, were explained by geographical location, whereas only 13.3% and 14.8% were explained by tree species identity. Additionally, 28.1%, 28.7%, and 26.7% of the bacterial variance in the root endosphere, rhizoplane, and rhizosphere, respectively, were explained by geographical location, while only 13.4%, 9.2%, and 16% were explained by tree species identity.

### 3.2. The Diversity Patterns

The cumulative number of OTUs (i.e., γ-diversity) for bacteria in the root-associated compartments (rhizosphere, rhizoplane, and root endosphere) exceeded that in leaf-associated compartments (leaf epiphytic and endophytic). Notably, γ-diversity was highest in the rhizosphere and rhizoplane among the five compartments, while it was lowest in the leaf endophytic (Figure 3A). Similarly, α-diversity (the observed OTU richness) was also observed to follow the same trend (Figure 3B). Multivariate analysis of variance further confirmed the influence of all factors considered in the experimental design. Results revealed that the geographic location significantly influenced the bacterial community α-diversity, whereas plant identity only affected that of the root endosphere (Appendix A). Further analysis unveiled that the observed OTU richness on the leaf epiphytic of *V. mangachapoi* was higher than in *D. pectinatum*, whereas the opposite was the root endosphere (Figure 3C). Moreover, higher OTU richness was observed in the leaf endophytic of *D. pectinatum* collected from Jianfeng, while OTU richness in the rhizosphere and root endosphere of *V. mangachapoi* collected from Diaoluo exceeded that of other locations (Figure 3D,E). In summary, our results revealed the significant influence of geographical location on bacterial community diversity.

### 3.3. The Impact of Environmental Factors

The variation partitioning analysis (VPA) was used to illustrate the contributions of species, climate, geographical location, and physicochemical properties of leaf and soil variables to bacterial community variation. Collectively, all variables explained 26% and 35% of the variation in leaf-associated and root-associated bacterial communities, respectively (Figure 4A,B). The VPA models revealed that factors exclusively linked to climate (*R*^2^ = 0.01) and geographical location (*R*^2^ = 0.01) rather than plant identity predicted a proportion of variation in leaf-associated bacterial communities (Figure 4A). Similarly, geographical location (*R*^2^ = 0.04) and soil physicochemical properties (*R*^2^ = 0.04) exclusively contributed more significantly to the variation in beta diversity of root-associated bacterial communities compared to plant identity (Figure 4B).

Further analysis revealed significant correlations between temperature (*R*^2^ = 0.22, *p* < 0.001), SOM (*R*^2^ = 0.13, *p* < 0.01), and TN (*R*^2^ = 0.21, *p* < 0.01) with leaf-associated bacterial communities (Figure 4C). Additionally, soil pH (*R*^2^ = 0.43, *p* < 0.001) and rainfall (*R*^2^ = 0.21, *p* < 0.001) exhibited significant correlations with root-associated bacterial communities (Figure 4D). It is noteworthy that redundancy analysis (RDA) also showed similar results (Appendix A). Linear regression analysis revealed the relationship between the individual environmental variables and the relative abundance of dominant bacterial classes, partially supporting these findings. TN (*R*^2^ = 0.59, *p* < 0.001) and SOM (*R*^2^ = 0.24, *p* < 0.001) were primarily associated with the relative abundance of *Actinobacteria* in leaf, while temperature (*R*^2^ = 0.29, *p* < 0.001) and pH (*R*^2^ = 0.30, *p* < 0.001) were primarily associated with the relative abundance of *Gammaproteobacteria* in leaf (Figure 5A,B). Moreover, TK (*R*^2^ = 0.19, *p* < 0.001) and soil pH (*R*^2^ = 0.36, *p* < 0.001) were mainly associated with the relative abundances of *Actinobacteria* and *Acidobacteriae* in the root, respectively (Figure 5C,D).

## 4. Discussion

### 4.1. Host Identity Was the Important Factor Affecting Plant Bacterial Community

The dominant taxa and biomarker taxa were considered potential keystone taxa, playing crucial ecological roles in microbiome assembly and ecosystem functions [41,42]. Biomarker analyses confirmed host selective mechanisms on bacterial community structure, evidenced by associations between dominant bacterial taxa and tree species (Appendix A). Notably, *Gammaproteobacteria* in *D. pectinatum* and *Actinobacteria* in *V. mangachapoi* emerged as the most significant biomarkers at the class level (Appendix A). This finding aligns with similar studies on other tree species [43,44,45]. Interestingly, *Gammaproteobacteria* and *Actinobacteria* were not co-marked in the same tree species, possibly due to the inhibitory effect of *Gammaproteobacteria* metabolites on *Actinobacteria* and host selection [46]. Moreover, *Gammaproteobacteria* contributed to nitrogen fixation and promoted the amplification of the relative abundance of functional genes [47,48,49,50,51]. Our findings confirm that major microbial taxa are sensitive to climate factors (temperature and rainfall) and nutrients, reflecting their adaptation to distinct environmental conditions [52,53]. Specifically, the relative abundance of *Actinobacteria* in the phyllosphere of *V. mangachapoi* surpassed that in *D. pectinatum*, with leaf nitrogen content being a key driving factor (Figure 5A). The structure of the leaf-associated bacterial community was associated with plant resource uptake strategies, such as leaf nitrogen content [54]. Studies indicated that long-term nitrogen application significantly increases the relative abundance of *Actinobacteria* in the soil, suggesting a close relationship between the growth of *Actinobacteria* and nitrogen content [55].

### 4.2. Geographic Location Was Another Main Factor Affecting Plant Bacterial Community

We also observed spatial differences in the relative abundance of dominant bacterial orders across different geographical locations (Appendix A). The non-random spatial variation in the relative abundance of these taxonomic units indicates a biogeographic pattern [56]. Geographic location accounted for most of the variation in environmental factors, compared to tree species identity (Appendix A). Similarly, geographic factors also explained more variation in bacterial communities (Figure 4A,B). Overall, the environmental differences shape the compositions of bacterial communities to some extent through environmental selection [57].

Specifically, the main driving factors of microbial communities, such as soil pH and climate factors (e.g., rainfall and temperature), were predominantly influenced by geographic locations, contributing to the mechanisms driving spatial variations in microorganisms. The environmental temperature was strongly associated with the leaf-associated bacterial community structure (Figure 4C,D), indicating that microorganism colonization is shaped by leaf-environment interactions [58]. This is unsurprising given that the phyllosphere presents a relatively open and nutrient-poor environment, subjecting microorganisms to various biotic and abiotic stresses [59]. The negative correlation between *Acidobacteria* and soil pH aligns with previous studies [60,61], consistent with our findings (Figure 5D). Furthermore, our results underscored the significance of soil pH in influencing root bacterial communities (Figure 4D), consistent with prior research [15,62,63]. In addition, *Gammaproteobacteria* of the phyllosphere exhibited a notable correlation with temperature variation across different geographical locations (Figure 5B). This could be explained by the predominance of *Gammaproteobacteria* species that respond to cold temperatures, leading to a significant decrease in their relative abundance with increasing temperatures [64,65].

### 4.3. Geographic Location Influenced More of the Bacterial Compositions than Host Species Identity

In conclusion, our study highlights the significant influence of both geographical location and tree species identity on bacterial community composition, with geographic location exerting a stronger effect. Our findings offer robust empirical evidence that geographic location accounts for a greater proportion of bacterial community variation in tropical trees compared to host species identity, even when considering tree species with distant genetic relationships (Figure 2B). This aligns with prior research suggesting that geographic location plays a more dominant role in shaping plant bacterial communities than tree species identity on broad spatial scales [66]. While this contradicts findings in temperate tree species, it may be attributed to distinct differences in bacterial composition between temperate and tropical tree species [26]. Notably, our results reveal that over 65% of bacterial composition variations remained unexplained by tree species, geographic location, or environmental variables. This could be due to the presence of other unmeasured environmental factors [56], including biological interactions such as competition, mutualism, and predation, among species [67].

## 5. Conclusions

This study primarily analyzed the community compositions of leaf and root-associated bacterial communities for the rare and endangered *D. pectinatum* and *V. mangachapoi* in tropical regions. Our findings indicated that geographic location explained more of the variation of bacterial communities than host species identity, even in tree species with distant genetic relationships. However, the driving factors of bacterial communities in the leaf and root differed. Leaf nitrogen content and temperature emerged as the primary drivers for the leaf-associated bacterial communities, while soil pH predominated as the driving factor for those in the root. These results expand our understanding of microbial community composition changes, particularly in tropical regions, and hold significant implications for advancing the conservation and sustainable management of endangered tropical tree species.

## Figures and Tables

**Figure 1 plants-13-01565-f001:**
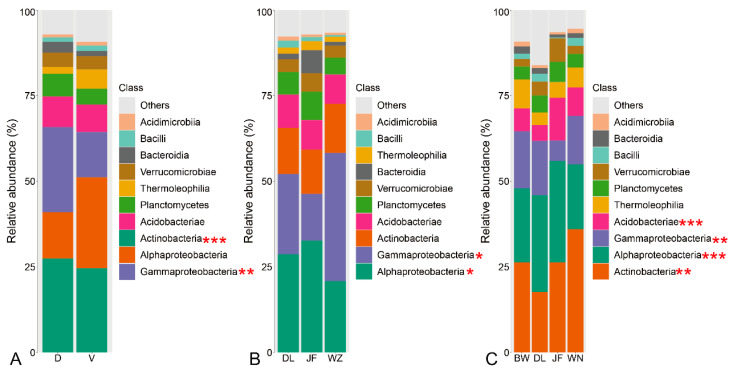
The bacterial community compositions (class level). (**A**) Different plants for *D. pectinatum* and *V. mangachapoi*. (**B**) *D. pectinatum* across three locations. (**C**) *V. mangachapoi* across four locations. Abbreviation: D, *D. pectinatum*; V, *V. mangachapoi*; DL, Diaoluo; JF, Jianfeng; WZ, Wuzhi; BW, Bawang; WN, Wanning. Significance level: * *p* < 0.05; ** *p* < 0.01; *** *p* < 0.001.

**Figure 2 plants-13-01565-f002:**
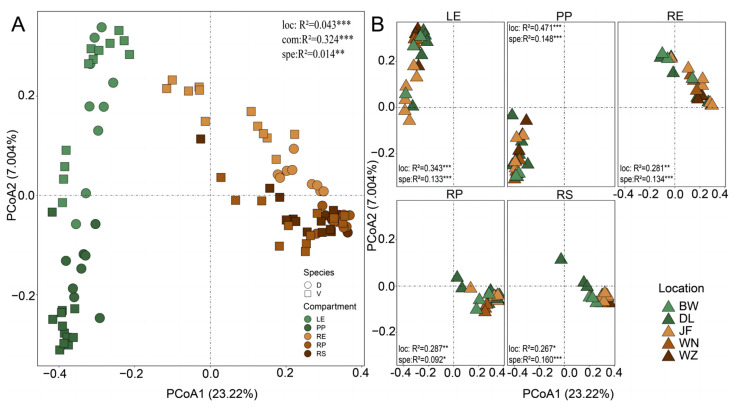
Principal coordinates analysis (PCoA) of taxonomic similarity based on Bray–Curtis distances for bacterial community compositions at the OTU level. (**A**) Compartment and species. (**B**) Geographical location and species. Significance level: * *p* < 0.05; ** *p* < 0.01; *** *p* < 0.001. Abbreviation: loc, geographical location; com, compartment; spe, species. LE, leaf endophytic; PP, leaf epiphytic; RP, Rhizoplane; RS, Rhizosphere; RE, root endosphere.

**Figure 3 plants-13-01565-f003:**
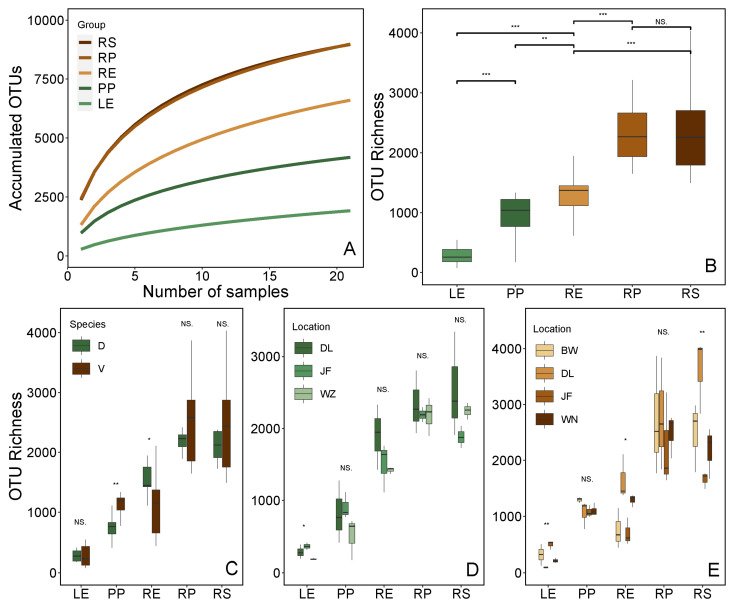
Diversity of the bacterial community. (**A**) Accumulated OTUs (γ-diversity) for all samples in five compartments; (**B**) Mean OTU richness (α-diversity) of five compartments. (**C**) Comparison of OTU richness of five compartments of *D. pectinatum* and *V. mangachapoi*. (**D**) Comparison of OTU richness of *D. pectinatum* across five compartments in three locations. (**E**) Comparison of OTU richness of *V. mangachapoi* across four compartments in four locations. Significance level: * *p* < 0.05; ** *p* < 0.01; *** *p* < 0.001. “NS” means *p* > 0.05, not significant.

**Figure 4 plants-13-01565-f004:**
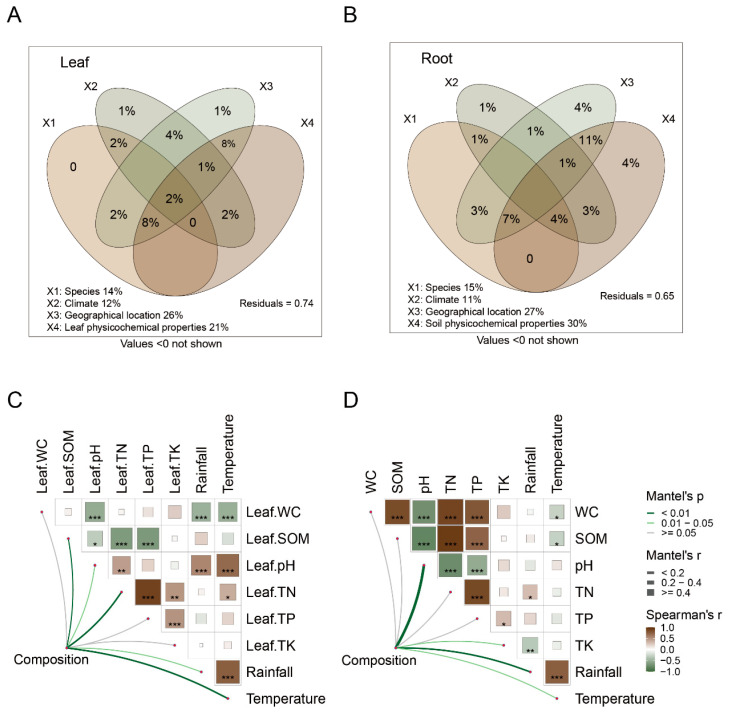
The relationship between environmental factors and bacterial communities in leaf and root. (**A**,**B**) Variation partitioning analysis (VPA) showing the effects of leaf and soil physicochemical properties, climate (rainfall and temperature), geographical location (latitude, longitude, and elevation), and species on bacterial community compositions (OTU level) in leaf and root. (**C**,**D**) Compositions of bacterial communities (OTU level) in leaf and roots are related to each environmental factor by partial Mantel tests. The edge width corresponds to Mantel’s r statistic for the corresponding distance correlation, while the edge color denotes the statistical significance based on 999 permutations. Significance level: * *p* < 0.05; ** *p* < 0.01; *** *p* < 0.001.

**Figure 5 plants-13-01565-f005:**
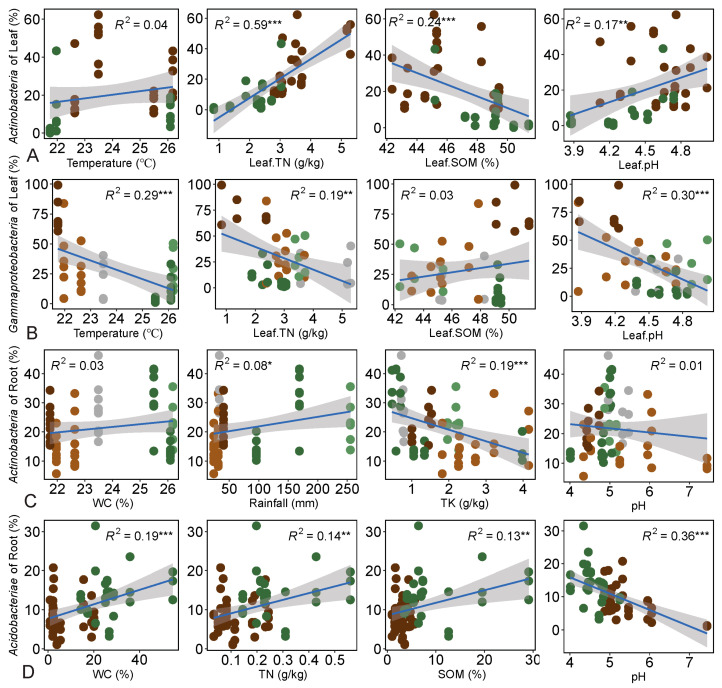
The relationship among environmental factors and the relative abundance of dominant bacterial class in leaf and root. (Only significant factors are displayed; *p* ≤ 0.05). (**A**) The relationship between environmental factors and *Actinobacteria* of the leaf. The dark green solid circles represent samples of *D. pectinatum*, and the dark brown is *V. mangachapoi*. The blue line and grey shade represent the fitted line and confidence interval, respectively. (**B**) The relationship between environmental factors and *Gammaproteobacteria* in leaf. The light green solid circles represent samples of Bawang, and dark green is Jianfeng. The light brown solid circles represent samples of Diaoluo, and dark brown is Wuzhi. The dark gray solid circles represent samples of Wanning. (**C**) The relationship between environmental factors and *Actinobacteria* in root. (**D**) The relationship between environmental factors and *Acidobacteriae* in root. Significance level: * *p* < 0.05; ** *p* < 0.01; *** *p* < 0.001.

## Data Availability

The raw reads were deposited into the NCBI Sequence Read Archive (SRA) database (Accession Number: PRJNA1085516).

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
