# Peer review of "Geographic Location Affects the Bacterial Community Composition and Diversity More than Species Identity for Tropical Tree Species"

_plants, 2024, doi:10.3390/plants13111565_

Round 1
Reviewer 1 Report
Comments and Suggestions for Authors
Dear Authors,
This study explored how plant species and location impact tropical plant bacterial communities. Findings show plant identity influences bacterial composition, while environmental factors and geographic location play significant roles in shaping these communities.
Majors:
The English is terrible, it needs a thorough revision, I'm not going to point out any specific mistakes because there are so many.
- The introduction is clearly deficient; it does not provide a good state of the art. It makes very obvious statements that are not necessary for the audience of a specialized journal like this one.
L44: “[6,7] There” There are many typographical errors, this is just an example, I'm not going to point out more, the authors should review the entire manuscript and correct them.
L57: “However, there are very few studies that have explored bacterial biogeogra-57 phy compared to fungi” There are many sentences that don't seem to make sense, like this one, for example. What's the point of talking about fungi.
L62: “phyllosphere” It hasn't been previously defined.
-L111-L115: These techniques should go in another section
-L143: “Data analysis” The section above is also data analysis
-L140-L142: They are results, not material
-L145-L146: It's redundant, it was already mentioned in the previous section.
-L191-L193: It's better to place this idea in the discussion section
-L220-L222: So it's the same conclusion as the previous section? It would be better to highlight what new this set of data contributes.
-L283: “Further, Gammaproteobacteria possessed formidable carbohydrate metabolism capabilities, which may promote the amplification of functional gene relative abundance” I'm sorry, I don't understand what the authors are trying to communicate with this idea? Could they please rewrite it?
-L285: “Meanwhile, Gammaproteobacteria in the phyllosphere have been identified as an important group of nitrogen-fixing bacteria” And what does this reference contribute to this research?
-I don't understand, this ideaL321: “Geographic location influenced more of the bacterial compositions than host species identity” Isn't it contrary to this other one? L273: “Host identity was the main factor affecting plant bacterial community”
- The format of the conclusions is not correct; they are not authentic conclusions. What the authors have done is another summary, along with some introduction. It is not appropriate to include bibliographic citations in the conclusions.
Minors:
L64: There are characters in Chinese
Comments on the Quality of English LanguageThe English needs a thorough revision
Author Response
Manuscript number: plants-3017573
Title: Geographic location affects the bacterial community composition and diversity more than species identity for tropical tree species
Journal title: Plants
Dear Reviewer,
I hope this email finds you well. I am writing to express my gratitude for your assistance in reviewing my manuscript titled “Geographic location affects the bacterial community composition and diversity more than species identity for tropical tree species” (plants-3017573). I sincerely appreciate the time and effort you and the expert reviewers have dedicated to providing invaluable feedback and suggestions.
After carefully considering the expert advice and suggestions, I have made significant revisions to the manuscript. I believe these changes have significantly strengthened the quality and clarity of the content. I have now re-uploaded the revised manuscript, which incorporates the recommended changes, and I would be grateful if you could kindly reassess it.
I want to take this opportunity to express my sincere appreciation for your guidance throughout this process. The valuable insights and recommendations provided by you and the expert reviewers have been instrumental in improving the overall quality of the manuscript. Your expertise and thorough evaluation have immensely contributed to the enhancement of my work.
Once again, thank you for your dedicated efforts and commitment to supporting my research. I am truly grateful for your assistance. Should you have any further suggestions or require additional information, please do not hesitate to let me know.
Looking forward to hearing from you.
Best regards,
Kepeng Ji
Reviewer 1
Comments and Suggestions for Authors
Majors:
The English is terrible, it needs a thorough revision, I'm not going to point out any specific mistakes because there are so many.
Response: Thank you for your valuable comments. We apologize for the problem. We have carefully revised the English expression in the article. Thanks again.
The introduction is clearly deficient; it does not provide a good state of the art. It makes very obvious statements that are not necessary for the audience of a specialized journal like this one.
Response: Thank you for your valuable comments. Revised. Thanks again.
L44: “[6,7] There” There are many typographical errors, this is just an example, I'm not going to point out more, the authors should review the entire manuscript and correct them.
Response: Thank you for your valuable comments. Revised.
L57: “However, there are very few studies that have explored bacterial biogeogra-57 phy compared to fungi” There are many sentences that don't seem to make sense, like this one, for example. What's the point of talking about fungi.
Response: Thank you for your valuable comments. After much thinking, as you said, it is not appropriate to talk about fungi here, because this study mainly focuses on bacteria. Therefore, we have deleted it. Thanks again.
L62: “phyllosphere” It hasn't been previously defined.
Response: Thank you for your valuable comments. We have revised the text in L45 to “One is the phyllosphere that mainly refers to the leaves, including epiphytic and endophytic microbial communities [8]. ” Thank you once again for bringing this to our attention.
L111-L115: These techniques should go in another section
Response: Thank you once again for bringing this to our attention. We appreciate your attention to detail. We have made an adjustment in the article according to your request.
L143: “Data analysis” The section above is also data analysis
Response: We appreciate your attention to detail. We have made an adjustment in the article according to your request.
L140-L142: They are results, not material
Response: Thank you once again for bringing this to our attention. We have made an adjustment in the article according to your request.
L145-L146: It's redundant, it was already mentioned in the previous section.
Response: Thank you for your valuable comments. We have deleted it. Thanks again.
L191-L193: It's better to place this idea in the discussion section
Response: Thank you for your valuable comments. Revised.
L220-L222: So it's the same conclusion as the previous section? It would be better to highlight what new this set of data contributes.
Response: Thank you for your valuable comments. Revised. These two conclusions are consistent. As can be seen from Table S2, geographical location significantly influenced bacterial community diversity. The previous section mainly studies the species composition of bacteria, while Figure 3 is mainly to explore the diversity of bacteria. Therefore, these are two parts of the study.
L283: “Further, Gammaproteobacteria possessed formidable carbohydrate metabolism capabilities, which may promote the amplification of functional gene relative abundance” I'm sorry, I don't understand what the authors are trying to communicate with this idea? Could they please rewrite it?
L285: “Meanwhile, Gammaproteobacteria in the phyllosphere have been identified as an important group of nitrogen-fixing bacteria” And what does this reference contribute to this research?
Response: Thank you for your valuable comments. There is consistency between the two questions you raised. Please allow me to explain them together. In the introduction, we emphasized the importance of plant microbiota and mentioned that one of our research goals was to investigate the bacterial communities of Dacrydium pectinatum and Vatica mangachapoi. Our results showed that Gammaproteobacteria was the dominant class in both. Therefore, we explained in the discussion that previous research results found the important functions of Gammaproteobacteria in order to express its importance. However, after considering your comments, we feel that this part is a bit cumbersome. Therefore, we have reorganized and clarified this part in the original text to ensure better understanding for readers. Thank you again for your feedback and informing us of any shortcomings.
I don't understand, this ideaL321: “Geographic location influenced more of the bacterial compositions than host species identity” Isn't it contrary to this other one? L273: “Host identity was the main factor affecting plant bacterial community”
Response: Thank you for your valuable comments. Please allow me to explain this comment. Our results showed that both geographical location and tree species identity had a significant impact on bacterial community composition (Figure 2). However, geographic location explained more of the variation of bacterial communities in tropical trees than host species identity (Figure 2). In other words, Geographic location influenced more of the bacterial compositions than host species identity. We guess there's no contradiction. Maybe our English expression is not accurate enough, we have changed "Host identity was the main factor affecting plant bacterial community" to "Host identity was the important factor affecting plant bacterial community".
The format of the conclusions is not correct; they are not authentic conclusions. What the authors have done is another summary, along with some introduction. It is not appropriate to include bibliographic citations in the conclusions.
Response: Thank you once again for bringing this to our attention. We have made an adjustment in the article according to your request.
Minors:
L64: There are characters in Chinese
Response: Sorry for the error. Revised. Once again, thank you for your valuable assistance in revising the details of our article. Your expertise and guidance have greatly contributed to improving the overall quality of our work. We have deleted it in our article.

Reviewer 2 Report
Comments and Suggestions for Authors
The authors performed a bacterial diversity study on Dacrydium pectinatum and Vatica mangachapoi at different locations on Hainan Island. I think this work is a good start and some important data is presented. However, I have some concerns about several aspects:
1) Sampling description is not very clear. How many trees were sampled in total? 3 per site? How many leaves in total were collected? Same thing about root samples. How many roots (or weight) in total? Was this the only sampling?
2) The authors used sonication to remove cells from leaves and roots. Do they prove empirically that the conditions used (40KHz for 1 minute; three times) did not lyse bacterial cells?
3) What is the distribution of the 11,841 OTUs obtained? How many are for each plant? How many are for leaves and roots?
4) Why the authors did not also study the Archaea present in the surface of these leaves and roots?
5) The manuscript should be revised by person fluent in English.
Comments on the Quality of English LanguageThe manuscript should be revised by person fluent in English.
Author Response
Manuscript number: plants-3017573
Title: Geographic location affects the bacterial community composition and diversity more than species identity for tropical tree species
Journal title: Plants
Dear Reviewer,
I hope this email finds you well. I am writing to express my gratitude for your assistance in reviewing my manuscript titled “Geographic location affects the bacterial community composition and diversity more than species identity for tropical tree species” (plants-3017573). I sincerely appreciate the time and effort you and the expert reviewers have dedicated to providing invaluable feedback and suggestions.
After carefully considering the expert advice and suggestions, I have made significant revisions to the manuscript. I believe these changes have significantly strengthened the quality and clarity of the content. I have now re-uploaded the revised manuscript, which incorporates the recommended changes, and I would be grateful if you could kindly reassess it.
I want to take this opportunity to express my sincere appreciation for your guidance throughout this process. The valuable insights and recommendations provided by you and the expert reviewers have been instrumental in improving the overall quality of the manuscript. Your expertise and thorough evaluation have immensely contributed to the enhancement of my work.
Once again, thank you for your dedicated efforts and commitment to supporting my research. I am truly grateful for your assistance. Should you have any further suggestions or require additional information, please do not hesitate to let me know.
Looking forward to hearing from you.
Best regards,
Kepeng Ji
Reviewer 2
Comments and Suggestions for Authors
Major remarks:
1) Sampling description is not very clear. How many trees were sampled in total? 3 per site? How many leaves in total were collected? Same thing about root samples. How many roots (or weight) in total? Was this the only sampling?
Response: Thank you for your valuable feedback on our manuscript " plants-3017573." We appreciate your comments and have carefully considered your suggestions. We have reorganized and clarified this part in the original text to ensure better understanding for readers. Thank you again for your feedback and informing us of any shortcomings.
We sampled a total of 21 trees, D. pectinatum at three sites and V. mangachapoi at four sites. Three trees were selected from each site, in other words, three replicates. Approximately 300 grams of mature and healthy leaf samples and 250 grams of root samples were taken from the same tree. To obtain epiphytic microbiota samples, microbial cells were dislodged and collected from 15–20 g of leaves and 10–15 g of roots. This is the only sampling.
2) The authors used sonication to remove cells from leaves and roots. Do they prove empirically that the conditions used (40KHz for 1 minute; three times) did not lyse bacterial cells?
Response: Thank you for your valuable feedback on our manuscript. We understand your concerns. It is your rigorous review that will make my paper better and better. The epiphytic microorganism extraction methods we used are all from references, such as 31 and 32 cited in my text, but we have made some improvements on this basis. Therefore, we believe that it is reliable and scientific. The sampling method used in this study is consistent with the method described in the previous article published by our team.
3) What is the distribution of the 11,841 OTUs obtained? How many are for each plant? How many are for leaves and roots?
Response: Thank you for your valuable comments. Please allow me to answer this for you. We feel so sorry, we have rechecked the data. In total, paired-end sequencing resulted in 3,808,575 high-quality reads, detecting 154,445 OTUs representing the bacterial community. There are 63782 OTUs in total for Dacrydium pectinatum, 9187 of which are related to leaves and 54595 to roots. Meanwhile, there are 90663 OTUs in total for Vatica mangachapoi, 16973 of which are related to leaves and 73690 to roots.
Sorry again for my oversight and it does not affect my final result. Regarding your question, the OTUs richness shown in Figure 3 of the manuscript clearly answers the distribution of OTUs. Figure 3B shows the distribution of OTUs in leaves and roots. The figure below shows the distribution of OTUs in each tree.
4) Why the authors did not also study the Archaea present in the surface of these leaves and roots?
Response: Thank you for your valuable comments. We did not pay attention to the archaea in leaves and roots, which is a pity. However, bacteria, as the largest group in plant leaves and roots, play a vital role. Therefore, in this study, we focused on bacteria. In the future, we will continue to conduct relevant research on plant archaea, and we believe that there will be some interesting discoveries.
5) The manuscript should be revised by person fluent in English.
Response: Thank you for your valuable feedback on our manuscript. We apologize for this and have made appropriate improvements.

Reviewer 3 Report
Comments and Suggestions for Authors
Dear Authors,
The manuscript plants-3017573 was submitted by Kepeng Ji et al. as Article to Plants journal and titled "Geographic location affects the bacterial community composition and diversity more than species identity for tropical tree species”.
Research object was microorganisms associated with plants bearing in mind that they play a vital role in their growth, development, and health. Research in this field is still really lacking. The authors found many interesting facts. The manuscript describes results that indicate that geographical locations had a greater impact than tree species identity on bacterial community compositions and diversity. The methods chosen for the research are valid and reliable. Extensive research yielded many results that are properly discussed. The findings are consistent with the results. Introduction is written very well. The sample collection strategy and methods are described in detail. The content, quality and design of the manuscript meet the requirements of the journal and the article can be accepted for publication after very minor revision.
Line 64, please delete 窗体顶端
Line 129, ddH2Oupto20μl. – spaces between words are missing.
Line 448, I believe that reference no. 36 has author, volume, pages, and year?
Sincerely,
2024-May-16
Author Response
Manuscript number: plants-3017573
Title: Geographic location affects the bacterial community composition and diversity more than species identity for tropical tree species
Journal title: Plants
Dear Reviewer,
I hope this email finds you well. I am writing to express my gratitude for your assistance in reviewing my manuscript titled “Geographic location affects the bacterial community composition and diversity more than species identity for tropical tree species” (plants-3017573). I sincerely appreciate the time and effort you and the expert reviewers have dedicated to providing invaluable feedback and suggestions.
After carefully considering the expert advice and suggestions, I have made significant revisions to the manuscript. I believe these changes have significantly strengthened the quality and clarity of the content. I have now re-uploaded the revised manuscript, which incorporates the recommended changes, and I would be grateful if you could kindly reassess it.
I want to take this opportunity to express my sincere appreciation for your guidance throughout this process. The valuable insights and recommendations provided by you and the expert reviewers have been instrumental in improving the overall quality of the manuscript. Your expertise and thorough evaluation have immensely contributed to the enhancement of my work.
Once again, thank you for your dedicated efforts and commitment to supporting my research. I am truly grateful for your assistance. Should you have any further suggestions or require additional information, please do not hesitate to let me know.
Looking forward to hearing from you.
Best regards,
Kepeng Ji
Reviewer 3
Comments and Suggestions for Authors
Line 64, please delete 窗体顶端.
Response: Thank you once again for bringing this to our attention. Sorry for the error. Revised.
Line 129, ddH2Oupto20μl. – spaces between words are missing.
Response: Sorry for the error. Revised. Once again, thank you for your valuable assistance in revising the details of our article.
Line 448, I believe that reference no. 36 has author, volume, pages, and year?
Response: Sorry for the error. Revised. Thank you for your valuable assistance in revising the details of our article.

Round 2
Reviewer 1 Report
Comments and Suggestions for Authors
Dear Authors,
I believe the authors have adequately addressed all of my comments and suggestions, and I accept the paper in its current version.
Reviewer 2 Report
Comments and Suggestions for Authors
My comments and suggestions were taken into account by the authors
Comments on the Quality of English LanguageMinor editing is required